# ProLLM: Protein Chain-of-Thoughts Enhanced LLM for Protein-Protein Interaction Prediction

**Mingyu Jin**[*]
Rutgers University

**Haochen Xue**[*]
University of Liverpool

**Zhenting Wang**
Rutgers University

**Boming Kang**
Peking University

**Ruosong Ye**
Rutgers University

**Kaixiong Zhou**
Massachusetts Institute of Technology

**Mengnan Du**
New Jersey Institute of Technology

**Yongfeng Zhang**[†]
Rutgers University

## Abstract

The prediction of protein-protein interactions (PPIs) is crucial for under-standing biological functions and diseases. Previous machine learning approaches to PPI prediction mainly focus on direct physical interactions, ignoring the broader context of nonphysical connections through interme-diate proteins, thus limiting their effectiveness. The emergence of Large Language Models (LLMs) provides a new opportunity for addressing this complex biological challenge. By transforming structured data into natural language prompts, we can map the relationships between proteins into texts. This approach allows LLMs to identify indirect connections between proteins, tracing the path from upstream to downstream. Therefore, we pro-pose a novel framework **ProLLM** that employs an LLM tailored for PPI for the first time. Specifically, we propose **Protein Chain of Thought (ProCoT)**, which replicates the biological mechanism of signaling pathways as natural language prompts. ProCoT considers a signaling pathway as a protein reasoning process, which starts from upstream proteins and passes through several intermediate proteins to transmit biological signals to downstream proteins. Thus, we can use ProCoT to predict the interaction between up-stream proteins and downstream proteins. The training of ProLLM employs the ProCoT format, which enhances the model's understanding of complex biological problems. In addition to ProCoT, this paper also contributes to the exploration of embedding replacement of protein sites in natural lan-guage prompts, and instruction fine-tuning in protein knowledge datasets. We demonstrate the efficacy of ProLLM through rigorous validation against benchmark datasets, showing significant improvement over existing meth-ods in terms of prediction accuracy and generalizability. Our results high-light the potential of LLMs to transform the field of PPI, serving as a robust potential tool for various categories of biological and medical research. The code is available at: https://github.com/MingyuJ666/ProLLM.

## 1 Introduction

Protein-protein interactions (PPIs) play an essential role in various biological processes of all living organisms, which are crucial for biomedical, genetic, and pharmaceutical research. Thus, numerous experimental methods have been proposed for PPI detection, such as yeast two-hybrid (Ito et al., 2001) and quantitative proteomics methods (Rotilio et al., 2012).

---

[*]Equal Contribution.

[†]Author Emails: mingyu.jin@rutgers.edu, hicaca945@gmail.com, zhenting.wang@rutgers.edu, kangbm@hsc.pku.edu.cn, ruosong.ye@rutgers.edu, kz34@mit.edu, mengnan.du@njit.edu, yongfeng.zhang@rutgers.edu

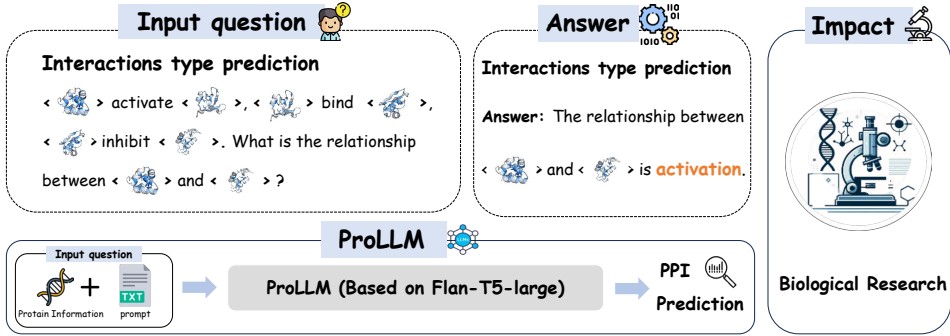

Figure 1: Illustration of the ProLLM Framework. We fine-tuning ProLLM under Human, SHS27K, SHS148K, and STRING datasets, enabling it to solve various PPI related tasks with the structure information purely described by natural language.

However, wet-lab methods for PPI prediction are often time-consuming and labor-intensive, highlighting the need for more precise and efficient computational tools.

In recent years, computational biology has developed rapidly. Methods such as the Convolutional Neural Network (CNN) and Graph Neural Network (GNN) have become powerful tools for studying protein interaction. CNN-based approaches like TAG-PPI (Song et al., 2022) typically use pre-trained embedding models to convert protein sequences into numerical vector representations, and then employ one dimensional convolutional neural networks to extract features from the vectors for subsequent PPI tasks.

Although CNN methods have shown some effectiveness in PPI prediction, they still have limitations due to their fixed receptive fields and the lack of well-defined spatial relationships in protein sequences, which limit the accuracy and interpretability of the predictions. GNN-based methods such as GNN-PPI (Lv et al., 2021) treat proteins as nodes and their relationships as edges, constructing a network composed of proteins, which better captures protein relationships and interactions, and outperforms CNNs in predicting protein interactions. However, while GNNs can effectively extract network structural information, they neglected the non-physical connections between two proteins without direct physical interactions, resulting in the worse performance in learning protein chains than transformer-based models (Zhou et al., 2020). Furthermore, GNNs cannot fully capture the relationships and dynamic changes in the signal passing process of living organisms, restricting their performance for PPI prediction (Zhou et al., 2023).

Following the GNN and CNN methods, Large Language Models have also been applied to this PPI area, such as ProBert (Elnaggar et al., 2021a) and ProteinLM (Xiao et al., 2021). As long as these models can obtain a protein representation, we can use the direct cosine similarity of the representation or train an MLP to perform PPI prediction. However, these methods still cannot capture the chain relationships between proteins, such as the signaling pathways. Besides, previous literature only used LLMs as a feature extractor. Recently, using LLM as a link predictor has shown that it can better capture relational information between nodes in knowledge graph tasks and its performance surpasses traditional GNN baselines (Ye et al., 2024; Zhuo et al., 2024; Shu et al., 2024). Therefore, it is promising to introduce LLM for protein-protein interaction (PPI) tasks, since the most important biological signal for PPI tasks is the chain relationships of proteins, i.e., the signaling pathways.

To bridge the gap, we propose **ProLLM**, with its key ideas illustrated in Figure 1, and the difference between the existing method and our ProLLM shown in Figure 2. Existing methods only focus on the single protein-protein interaction, overlooking the application of protein links to predict PPI in signaling pathways. Instead, we employ a large language model to learn the law of signal pathways and adapt the LLM to directly predict the type of interaction between proteins.

The signaling pathway addresses the traditional method's ignorance of global, non-physical connections between proteins. Signaling pathways typically start with an upstream protein that sends a biological signal through several intermediates to a downstream protein, hence

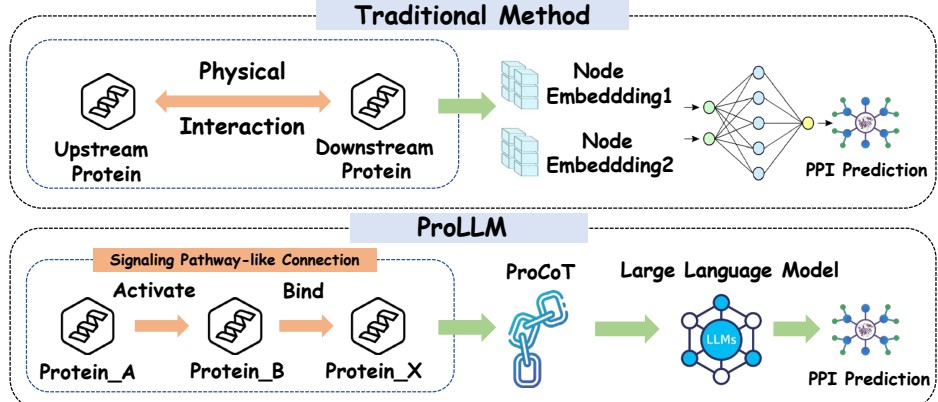

Figure 2: The difference between the existing method and our method in PPI prediction. Existing method focus on the property of upstream protein and downstream protein, our method focus on signaling pathway-like connection.

requiring consideration of the cumulative effect of multiple protein interactions. This series of interactions form sequential chains. Therefore, we propose **Protein Chain of Thought (ProCoT)** to overcome the limitation in understanding signaling pathways and protein functions. ProCoT is a data format that simulates the signal transduction process using a thought-chain approach, thereby enabling the prediction of protein interactions in signaling pathway problems. CoT can express the thinking process step by step to form a reasoning chain (Jin et al., 2024b), while ProCoT extends this principle further into the protein-related domain to simulate protein signaling pathways, giving LLMs a deeper insight into protein.

Additionally, our approach addresses the issue of poor protein comprehension in LLMs by replacing the standard language model embedding with embedding infused with protein information. When we process the protein name in the prompt, we replace its original embedding by ProtTrans (Elnaggar et al., 2021b), because its embedding contains protein's structual information. We also perform the instruction fine-tuning on the protein knowledge dataset to infuse protein domain knowledge into LLMs. Following these steps, the LLM acquires a robust ability to reason about the direct relationships between proteins, as demonstrated in Figure 1. It can provide answers to questions about protein relationships, which play a significant role in biological research.

Our contributions are summarized as follows:

- To the best of our knowledge, we are the first to explore the PPI prediction problem as a natural language processing problem. The proposed Protein Chain of Thought (ProCoT) is a novel method for understanding complex multi-step protein interactions, i.e., those present in signaling pathways.

- We propose embedding replacement and instruction fine-tuning on our model to enhance its ability to understand and reason about proteins. This also provides our model with rich background knowledge of protein sequences and protein interactions before training.

- Experiments on four widely used PPI datasets (i.e., Human, SHS27K, SHS148K, and STRING) demonstrate that ProLLM outperforms graph-based methods such as GNN-PPI and SemiGNN-PPI. It also has better performance than LLM-based methods like Instruct-GLM. The micro-F1 scores on these 4 datasets are 91.05, 78.09, 87.66, 89.21, respectively.

## 2 Related Work

**Protein-protein Iteractions**  Protein-protein interactions (PPIs) are indispensable for all living organisms, and so many efforts have been made for PPI detection up to now. Yeast two-hybrid (Y2H) assays (Brückner et al., 2009), synthetic lethality (O'Neil et al., 2017), quantitative proteomics (Wilm, 2009), and mass spectrometry (Mann et al., 2001), are widely used for identifying PPIs. To be specific, Y2H assays explore binary PPIs in living cells, offer-

ing insight into protein functions and regulations, although labor-intensive and limited in genomic coverage. Synthetic lethality identifies essential gene pair interactions by revealing compensatory relationships when both are inactivated. Meanwhile, quantitative proteomics, especially through mass spectrometry, illuminates the dynamic nature of the interactome under various conditions. Although informative, these methods require significant labor, time, and resources.

**Traditional Machine Learning Models for PPI Prediction**    In the realm of machine learning, sequence-based approaches include Shen's SVM method (Shen et al., 2007), which uses a 3-mer count vector from protein sequences as features and groups 20 amino acids into seven classes to handle synonymous mutations and reduce feature space dimensionality. SVM-based methods (Guo et al., 2008) and the ensemble model PCA-EELM (Principal Component Analysis-Ensemble Extreme Learning Machine) (You et al., 2013) utilize various types of protein sequence information for PPI prediction. In the domain of deep learning, DeepPPI (Du et al., 2017) extracts a multitude of features from protein sequences and employs a dual deep neural network structure for feature fusion and prediction. Sun et al. (Sun et al., 2017) introduced a PPI predictor based on a stacked autoencoder, emphasizing the importance of sample balance. DPPI (Hashemifar et al., 2018) and TAGPPI (Song et al., 2022) further extend the application of convolutional neural networks and integrate text convolution networks with graph representation learning to enhance the accuracy of PPI predictions. GNNs have significantly advanced PPI predictions, improving our understanding of biological mechanisms. GNN-PPI (Lv et al., 2021) enhances inter-novel-protein prediction accuracy by utilizing protein relationships and a new evaluation approach. PT-GNN (Long et al., 2022) integrates diverse data for link prediction, learning node features from sequence and structure. DeepRank-GNN (Réau et al., 2023) offers a modular, Python-packaged framework for GNN-based interaction pattern predictions. HIGH-PPI (Gao et al., 2023) introduces a hierarchical graph learning model for effective PPI prediction and molecular detail extrapolation. Geometric GNNs excel in modeling spatial intricacies, enhancing biomolecule prediction accuracy. Geo-PPI (Liu et al., 2021) utilizes self-supervised learning for geometric protein structure representations, excelling in detailing protein interactions. mmCSM-PPI (Rodrigues et al., 2021) captures multifaceted features for mutation impact predictions on protein interactions. MAPE-PPI (Wu et al., 2024) defines the microenvironment of amino acid residues in proteins and encodes it into discrete codes, which can capture the connections between different microenvironments, enhancing the prediction accuracy of PPI.

**Large Language Model for PPI prediction**    Recent advances in large language models, such as BERT (Devlin et al., 2018), GPT (Peng et al., 2023), LLaMA (Touvron et al., 2023), and T5 (Raffel et al., 2020), have significantly advanced the field of Natural Language Processing (NLP) to new heights. These models, having been trained on extensive textual corpora, exhibit exceptional capabilities across a diverse range of NLP applications (Shengyuan et al., 2024; Jin et al., 2024a; Fan et al., 2024; Hua et al., 2024). Inspired by LLMs, Protein Large Language Models(PLMs) pre-trained on large-scale protein sequences have emerged, such as ESM (Hsu et al., 2022), ProtTrans (Elnaggar et al., 2021b) and ProteinBert (Elnaggar et al., 2021b). PLMs provide a better representation of protein sequences by converting the protein sequences into the form of high-dimensional numerical vectors, known as embedding. With the protein sequences captured by the PLMs, the performances on diverse downstream tasks, such as structure prediction (Lin et al., 2023), subcellular localization prediction (Thumuluri et al., 2022), single peptide prediction (Teufel et al., 2022) and N-linked glycosylation sites prediction (Hou et al., 2023), have been transformed. It can be expected that PLMs will assist in PPI prediction tasks. ProtLLM (Zhuo et al., 2024) utilizes a dynamic protein mounting mechanism, a protein-as-word language modeling approach, and the InterPT dataset for pre-training, enabling it to handle complex inputs and achieve superior performance on various proteins-related tasks. However, ProtLLM is used for general protein tasks, and it is not used for PPI tasks exactly. The methodology of training these LLMs to convert text inputs to desired text outputs positions them as particularly advantageous for tasks such as generative link prediction (Ye et al., 2024; Shu et al., 2024). In such tasks, the model is tasked with inferring and generating the relationship between two entities based on provided textual cues. Moreover, the extensive pre-training phase

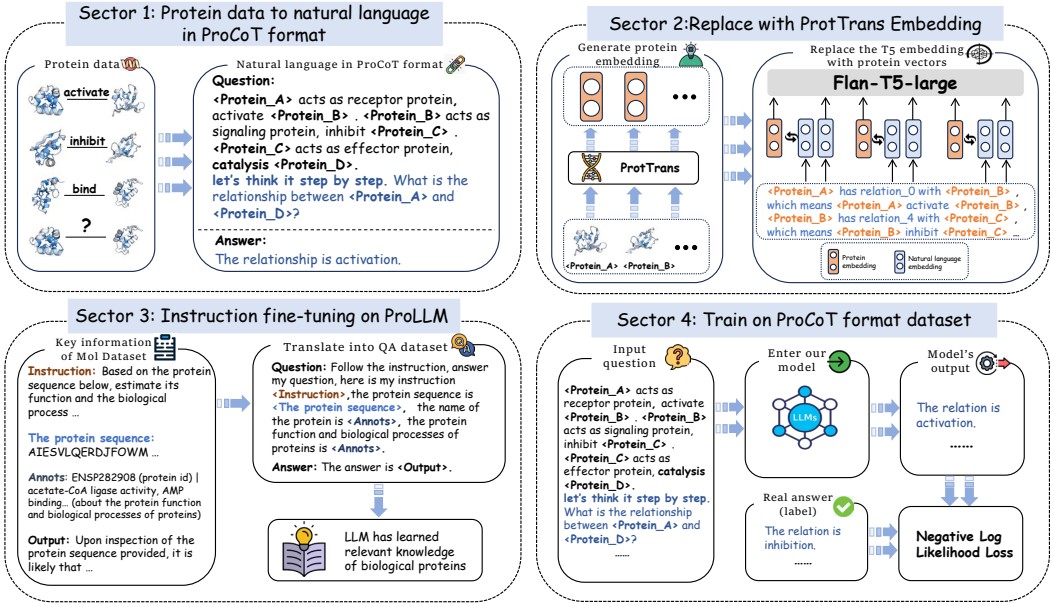

Figure 3: The process of ProLLM. Sector 1: Transfer the original protein data into ProCoT format of natural language that indicates the signaling pathways between proteins; Sector 2: Replace protein information embeddings with natural language embeddings to enhance the model's understanding of proteins; Sector 3: Inject knowledge about protein function; Sector 4: Fine-tuning on the ProCoT format dataset in Sector 1.

enables LLMs to exhibit a remarkable capacity for generalization. This capacity allows them to effectively tackle and respond to tasks or prompts that were not explicitly covered during their initial training (Wei et al., 2022). In addition to the outlined capabilities, the inherent flexibility and generalization potential of LLMs suggests that their applicability extends well beyond the conventional boundaries of NLP tasks (Yang et al., 2023). Specifically, their proficiency in generalizing from expansive pre-training sessions paves the way for their application in fields like bioinformatics and complex network analysis.

## 3 Proposed ProLLM Framework

In this section, we will introduce the implementation details of **ProLLM**, a framework designed to transform protein interaction data into ProCoT format natural language descriptions to simulate protein signaling pathways. By translating the structure relationships of proteins into natural language, we effectively transform protein interaction problems into natural language processing (NLP) tasks. To enhance the understanding of proteins by ProLLM, we directly integrate the protein vectors generated from ProtTrans (Elnaggar et al., 2021b) into our model to replace the original word embedding in the protein name's place. This approach allows our model to understand and utilize the biological attributes of proteins when predicting protein interactions. The ProLLM process is shown in Figure 3.

### 3.1 ProCoT

#### 3.1.1 Protein Data to Natural Language in CoT Format

Transformation of the original protein data into a ProCoT natural language is a critical step, this kind of natural language type that represents the relationships of proteins is shown as Definition 1.

**Definition 1** *(Protein Interaction) Given the set $\mathcal{P}$ representing all proteins under consideration in the study, we define the interaction between any two proteins $p_n, p_m \in \mathcal{P}$ with the specific type interaction $R_i$ interaction set $\mathcal{R}$. This relationship is encapsulated as a tuple $(p_n, p_m, R_i)$.*

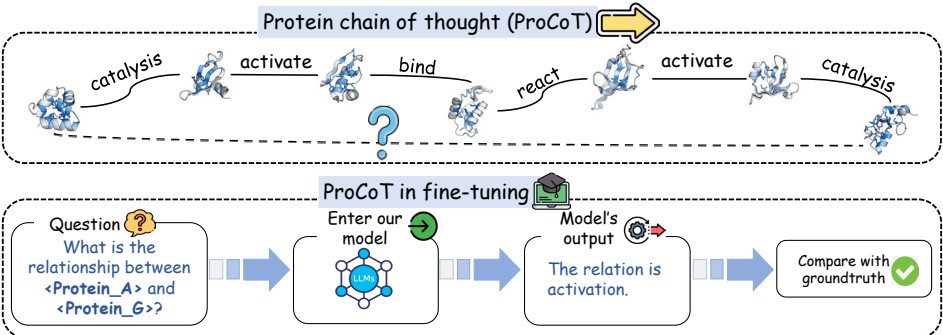

Figure 4: The fine-tuning process of ProCoT. Within the first dashed box, solid lines between proteins represent the signaling pathway, and the dashed lines connecting the head and tail proteins indicate the masked interaction. Our model will predict the type of masked interaction.

**Definition 2** *(Protein Signaling Pathway) In cellular signal transduction, receptor protein $\mathcal{R}c$ is the beginning of the signal pathway, and it can interact with downstream signaling proteins $\mathcal{S}p$. $\mathcal{S}p$ is responsible for continuing to transmit signals to effector protein $\mathcal{E}p$ inside the cell. After the $\mathcal{E}p$ response to the signal, it will trigger specific biological effects and activate secondary messengers $\mathcal{M}$. $\mathcal{M}$ may be the end of this signal level, or it may open a second signal hierarchy. If in the second signal level, the $\mathcal{M}$ will replace the $\mathcal{R}c$ in the initial signal hierarchy as the starting point. The signal will continue to propagate from $\mathcal{R}c$ through new signaling proteins, and effector proteins, and then generate new secondary messengers. After this stage, the propagation of the signal repeats the structure of the second signal level until the signal is interrupted.*

We aim to improve the model's understanding of biological signaling pathways, enhancing its ability to learn and reason interactions within complex signal networks. We create prompts in the ProCoT (Protein Chain of Thought) format to incrementally decompose the signal transduction process like Definition 2, simulating real pathways of signal propagation. Specifically, we clarify the structure knowledge within the signaling pathway, such as signaling proteins and effector proteins. We then design rules for signal transmission across different levels to simulate the iterative process of signal transduction in proteins. Finally, we also design signal interruptions to simulate the continuity of protein transmission in real life. We use hard code to convert protein interaction information as Definition 1 in dataset into ProCoT format natural language to imitate protein signaling pathways as Definition 2. Figure 7 in the appendix is an example of ProCoT. The answer to this is "The relationship is activation".

### 3.1.2 Training on ProCot Format Dataset

In biology, co-expression and co-localization refer to the phenomenon in which proteins that are often expressed or located together in the cell tend to participate in the same or interconnected biological processes (Zhang et al., 2019). Thus, biologists frequently use known protein information to infer unknown protein interactions. Based on the principles of co-expression and co-localization in biology, we have formulated our training strategy. After constructing our ProCoT training data using DFS, we will obtain a circular protein interaction as in Figure 4. We mask the relationship between the initial and final proteins in the signaling pathway and let our model predict this relationship.

### 3.1.3 Why ProCot Works: Biological Intuition behind ProCot

In this section, we will explore ProCoT through the lenses of biology and AI. **(1) Simulating Biological Signaling Pathways:** The biological signaling pathway is a series of ordered, rule-based interaction processes. During the process, the output of each step serves as the input for the next step, forming a highly organized information transmission chain. This pattern is consistent with the methods of handling serialized information in natural language processing (NLP). Additionally, Flan-T5's self-attention blocks can capture the indirect relationships between proteins that are distant and do not have direct interactions.

It is crucial for understanding biological signal transduction processes that involve multiple steps and complex intermediate links. **(2) Signal Interruption Mechanism:** This mechanism aims to mimic protein adaptability since the signaling between proteins needs to be constantly interrupted to ensure the cell's accurate response to external changes. This mechanism we designed can satisfy the complex feedback mechanisms and regulatory networks within the cell. Overall, our ProCoT design follows biological principles.

## 3.2 Enhancing Protein Sequence and Function Comprehension

In our methodology, we enhance the model's understanding of protein sequence by replacing the T5 embedding with the ProTrans embedding vectors and also perform instruction fine-tuning to enable the model to learn protein functions.

### 3.2.1 Embedding replacement

We use a new embedding mechanism facilitated by the `ProTrans` (Elnaggar et al., 2021a) model to perform embedding replacement. `ProTrans` is a large-scale protein model capable of transforming protein sequences into embedding vectors with the biophysical and structural features of proteins.

**Definition 3** *(Embedding Replacement) Given the protein id $P_{id}$, we can query its protein sequence $P_{seq}$ by the professional bioinformatics tool Ensembl BioMart. $P_{seq}$ will be the input of ProTrans and ProTrans will output a $1 \times 1024$-dimensional vector $P_{emb}$.*

We add all protein IDs in the protein dataset and their corresponding embeddings to the T5 vocabulary to solve the OOV (out of vocabulary) problem: The T5 vocabulary does not contain words for protein ID numbers. Because ProtTrans is trained based on T5-large, the process of adding to the vocabulary does not involve any change in dimensions, and there is no information loss. Additionally, this approach can also avoid the issue where a protein ID is tokenized into multiple tokens, causing the model to fail to understand it.

The replaced embedding vectors from ProtTrans can add a lot of prior knowledge about the intrinsic patterns and biophysical features of proteins to our model, better applying it to subsequent protein interaction prediction tasks.

### 3.2.2 Instruction Finetuning

Mol-Instructions (Mol) (Fang et al., 2023) dataset is applied to conduct instruction fine-tuning on our model. Mol is a comprehensive instruction dataset designed for the field of biomolecules that aims to address the limitations of LLMs in biomolecular research. We use Mol to teach our model with knowledge of protein functions.

**Definition 4** *(Description of Mol) The content of the Mol dataset can be defined as follows: instructional text I related to LLMs queries, an input amino acid sequence S that includes essential protein information, and metadata M that sheds light on vital details like the protein's subcellular location, its primary function, and its participation in biological processes, followed by a corresponding output O that serves as the response to I.*

To facilitate instruction-based fine-tuning, we convert these objects into prompt-answer pairs $(P, A)$. This is shown as Figure 8 in the appendix. We convert the entire Mol dataset into a prompt-answer format and using these prompt-answer pairs for instruction fine-tuning of our model(ProLLM). This part can enhance the model's predictive performance on protein-related tasks.

## 4 Experiments

In this section, we present the experimental results to answer the following research questions (**RQ**s).

- **RQ1** - How does the performance of the proposed ProLLM framework compare to other baselines in terms of PPI prediction accuracy and generalizability?
- **RQ2** - How do different LLM backbones (e.g., Flan-T5-base and LLaMA-7b) affect the performance of the proposed ProLLM framework?
- **RQ3** - What are the contributions of each component to the PPI prediction performance of the ProLLM framework?

## 4.1 Experimental Settings

**Datasets:**   We undertake comprehensive evaluations on a trio of public Protein-Protein Interaction (PPI) datasets: Human (Song et al., 2022), STRING (Szklarczyk et al., 2019), SHS27k, and SHS148k (Chen et al., 2019). In these four datasets, we employ DFS dataset partitioning techniques. By prioritizing depth, DFS can effectively capture the step-by-step signal transmission and the hierarchical signal level of protein signaling pathways. Each dataset is split into training, validation, and testing sets, maintaining a proportion of 70%, 10%, and 20% respectively.

**Baselines:**   Earlier studies on predicting PPI are not pre-trained, they do not have prior knowledge about the proteins. Hence, we choose SVM (Romero-Molina et al., 2019), DPPI (Hashemifar et al., 2018), DNN-PPI (Li et al., 2018), PIPR (Chen et al., 2019), GNN-PPI (Lv et al., 2021) and SemiGNN-PPI (Zhao et al., 2023) as the baseline without pre-training. Furthermore, we explore how protein pre-training can influence the PPI, we include ProBERT (Elnaggar et al., 2021a), SM-1b (Rives et al., 2021), GearNet-Edge (Zhang et al., 2022), and KeAP (Zhou et al., 2022) as pre-trained baselines. MAPE-PPI (Wu et al., 2024) and InstrucGLM (Ye et al., 2024) have two versions: one with pre-training and one without pre-training.

**Evaluation Metrics:**   We selected the micro-F1 score (Harbecke et al., 2022) as our evaluation metric because the PPI dataset exhibits class imbalance, making the F1 score a very relevant reference. Note that micro-F1 is widely used in the protein-protein interaction task (Tran & Kavuluru, 2018). Each dataset follows a 70% training, 10% validation, and 20% testing data split. Subsequently, we will choose different random seeds for training and testing, conducting a total of 10 tests. The mean micro-F1 score across these trials will serve as the definitive measure of model performance, with the accompanying standard deviation reflecting variability in different experimental runs.

## 4.2 Comparative Experiment (RQ1)

We compare the performance of ProLLM with other baselines (w/ and w/o additional pre-training data) on four datasets in Table 1. Note that here we use Flan-T5-large (Raffel et al., 2020) as the backbone model. Based on the results, we can make three important observations: (1) Our method outperforms other baselines without pre-training. Although InstructGLM is also LLM based, it lags behind ProLLM; (2) As for the models pre-trained on protein dataset, they cannot achieve the performance of ProLLM without prior knowledge; (3) ProLLM outperforms GearNet-Edge, KeAP, and MAPE-PPI, although they utilize a significantly larger dataset comprised of structural and knowledge graph data for pre-training than the Mol dataset.

## 4.3 Influence of Different Backbones (RQ2)

We choose Flan-T5-base, Flan-T5-large, Flan-T5-XL (Raffel et al., 2020) and LLaMA-7b (Touvron et al., 2023) models as the backbone for ProLLM. We report the micro-F1 performance comparison as in Table 2. We should replace the embedding in Flan-T5 with ProtTrans and ProtTrans is a pre-trained model based on Flan-T5-Large. Therefore, the embedding generated by ProtTrans will match Flan-T5-large better. Additionally, despite having more model parameters, LLaMA-v1-7b exhibits worse lower micro-F1 scores in four datasets compared to lighter models: Flan-T5-base, Flan-T5-large, and Flan-T5-XL. Furthermore, the greater standard deviation of LLaMA-v1-7b highlights its instability.

| Method | Pre-training Dataset | Human | SHS27k | SHS148k | STRING |
|---|---|---|---|---|---|
| SVM | - | 61.28±1.28 | 53.07±5.16 | 58.59±0.07 | 64.59±0.03 |
| DPPI | - | 54.19±0.78 | 46.12±3.02 | 52.03±1.18 | 66.82±0.29 |
| DNN-PPI | - | 61.72±1.30 | 54.34±1.30 | 58.42±2.05 | 64.94±0.93 |
| PIPR | - | 62.72±0.50 | 57.80±3.24 | 63.98±0.76 | 67.45±0.30 |
| GNN-PPI | - | 78.61±1.38 | 66.52±5.26 | 75.34±1.54 | 84.28±0.89 |
| SemiGNN-PPI | - | 80.79±1.40 | 69.25±3.91 | 77.62±1.08 | 84.85±0.65 |
| MAPE-PPI | - | 82.13±1.47 | 72.04±3.46 | 80.45±1.12 | 86.48±0.52 |
| InstructGLM | - | 81.35±2.04 | 70.01±3.75 | 75.35±1.98 | 84.15±1.85 |
| **ProLLM(Flan-T5-large)** | - | **87.32±1.93** | **75.13±3.76** | **85.13±1.86** | **87.12±1.68** |
| ProBERT | BFD | 79.58±0.76 | 68.85±3.18 | 74.76±1.21 | 83.82±0.49 |
| EMS-1b | UniRef50 | 81.48±1.02 | 70.69±3.40 | 79.64±1.93 | 85.21±0.76 |
| KeAP | ProteinKG25 | 82.31±0.71 | 72.38±2.96 | 80.20±1.26 | 86.58±0.41 |
| GearNet-Edge | AlphaFoldDB | 82.87±1.02 | 72.06±3.56 | 79.84±1.65 | 85.96±1.01 |
| MAPE-PPI | CATH4.2 | 83.64±1.22 | 73.21±2.97 | 81.78±1.24 | 87.23±0.35 |
| InstructGLM | Mol dataset | 85.71±2.01 | 75.64±3.49 | 83.41±1.78 | 85.25±1.72 |
| **ProLLM(Flan-T5-large)** | Mol dataset | **91.05±1.63** | **78.09±3.24** | **87.66±1.68** | **89.21±1.45** |

Table 1: The micro-F1 of different methods (w/o and w/ additional pre-training data) on different datasets, where bold and underline denote the best and second best metrics, respectively. Higher micro-F1 denotes better performance.

| Method | Pre-training Dataset | Human | SHS27k | SHS148k | STRING |
|---|---|---|---|---|---|
| ProLLM-Flan-T5-base | - | 82.62±2.01 | 71.67±4.36 | 78.19±1.82 | 80.63±1.97 |
| **ProLLM-Flan-T5-large** | - | **87.32±1.93** | **75.13±3.76** | **85.13±1.86** | **87.12±1.68** |
| ProLLM-Flan-T5-XL | - | 84.32±2.65 | 73.28±3.96 | 81.73±2.21 | 82.94±1.93 |
| ProLLM-LLaMA-v1-7b | - | 81.75±4.46 | 70.33±6.52 | 77.08±3.59 | 79.41±2.87 |
| ProLLM-Flan-T5-base | Mol dataset | 87.92±2.07 | 74.28±4.09 | 85.07±1.87 | 87.11±1.88 |
| **ProLLM-Flan-T5-large** | Mol dataset | **91.05±1.63** | **78.09±3.24** | **87.66±1.93** | **89.21±1.45** |
| ProLLM-Flan-T5-XL | Mol dataset | 89.16±2.41 | 75.96±3.38 | 86.13±1.97 | 87.97±1.78 |
| ProLLM-LLaMA-v1-7b | Mol dataset | 87.08±4.72 | 74.87±6.51 | 84.61±3.53 | 86.71±2.61 |

Table 2: The micro-F1 score of ProLLM on different backbones. Higher micro-F1 denotes better performance. Bold and underline denote the best and second-best metrics, respectively.

## 4.4 Ablation Study (RQ3)

In our ablation study (Table 3), we evaluated the impact of different configurations on PPI prediction. The configurations are as follows: **ProLLM w/o ProCoT:** This setup shuffles ProCoT data, disrupting signaling pathways to mimic the model's performance without pathway understanding. It limits the model's ability to learn from signaling sequences. ProLLM relies on memorizing fixed relations instead of reasoning through intermediate protein relationships; **ProLLM w/o Embedding Replacement:** It will compare the model with replaced embeddings to the model without expended vocabulary, which evaluates the effect of protein-specific embedding features; **ProLLM w/o Instruction Fine-tuning:** This setup examines the model's capability to predict protein-protein interactions without the application of instruction fine-tuning on Mol dataset; **ProLLM w/o Embedding and Instruction Fine-tuning:** This configuration tests the model's performance without utilizing both protein-specific embedding features and instruction fine-tuning on Mol dataset.

In our ablation study, we have identified that ProCoT has the most significant impact on the performance of ProLLM. Our experiments revealed that introducing ProCoT led to substantial improvements in the performance of the model. Additionally, we explored other techniques such as embedding replacement and instruction fine-tuning on the Mol dataset. While these approaches did show some positive effects on the model's performance, their impact was found to be comparatively smaller when compared to the influence of ProCoT.

## 5 Conclusions and Future Work

In this paper, we propose ProLLM, a novel framework that leverages LLMs for protein-protein interaction prediction by representing protein data in natural language formats. Our key contributions include: 1) ProCoT (Protein Chain of Thought) to convert multi-step protein signaling pathways to natural language prompts, and the design of ProCoT can reflect the actual protein signaling passing within a biological organism. Additionally,

| ProCoT | Embedding Replacement | Instruction Fine-tuning | Dataset | | | |
|:---:|:---:|:---:|:---:|:---:|:---:|:---:|
| | | | Human | SHS27k | SHS148k | STRING |
| ✓ | | | 83.87±1.25 | 70.83±1.54 | 79.12±2.64 | 82.68±1.36 |
| ✓ | ✓ | | 87.32±1.93 | 74.26±3.53 | 85.13±1.86 | 87.12±1.68 |
| ✓ | | ✓ | 88.53±1.45 | 73.59±2.29 | 85.94±3.39 | 87.62±1.68 |
| | ✓ | ✓ | 78.32±2.65 | 61.98±2.06 | 74.10±1.27 | 77.85±1.54 |
| ✓ | ✓ | ✓ | **91.03±1.63** | **78.09±3.24** | **87.64±1.93** | **89.28±1.45** |

Table 3: Ablation study. The metric here is micro-F1. Where bold denote the best metrics. Higher micro-F1 denotes better performance.

the format of ProCoT is sequential, which is a type of information that LLMs are good at processing. 2) Integration of protein-specific embeddings from ProtTrans, and 3) Instruction fine-tuning on protein knowledge datasets. Through extensive experiments on four PPI datasets, ProLLM significantly outperformed existing graph-based and language model methods in prediction accuracy and generalizability. By unifying language models with structured biological data, our work opens up new possibilities for driving discoveries in computational biology, drug discovery, and broader scientific domains.

## 5.1 Acknowledgement

We thank Wenyue Hua, Kai Mei and Taowen Wang for their valuable discussions and suggestions during the project.

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

# 6 Appendix

## 6.1 Dataset Partition Algorithm

In the field of graph learning, DFS (Depth-First Search), BFS (Breadth-First Search), and random sampling are three common graph traversal or sampling strategies. Figure 5 shows the difference in between DFS and BFS.

**Depth-First Search (DFS):** DFS starts from a starting node and explores the graph's depth until it cannot go further, then backtracks to the nearest unvisited node from the starting node. This approach makes DFS inclined to explore the deep structure of the graph. **Breadth-First Search (BFS):** BFS starts from a starting node and visits its neighboring nodes one by one, then proceeds to visit the neighboring nodes' neighboring nodes, and so on. This approach prioritizes exploring the breadth of the graph. **Random Sampling:** Random sampling is a method of randomly selecting nodes for traversal or sampling. It can employ uniform random selection or select nodes based on certain probabilities.

The choice of strategy depends on the specific problem requirements and DFS is for exploring entire connected components. DFS can be used to find paths in a graph, especially when finding all paths from one node to another. DFS selects the next node for in-depth exploration at each step until the target node is found or cannot continue deeper. This is very similar to the protein signaling pathway in biology. From one protein to the target protein through different proteins, to simulate the signaling pathway. Additionally, after the DFS, we can obtain a cyclic structure of connected proteins, where one side of the cycle represents the signaling pathway between all proteins from the head protein to the tail protein, and the other side represents the direct interaction between the head and tail proteins. This is the data format we need in training our model. To simulate signaling pathways for training, we propose ProCot, and we **only use DFS** for dataset partition.

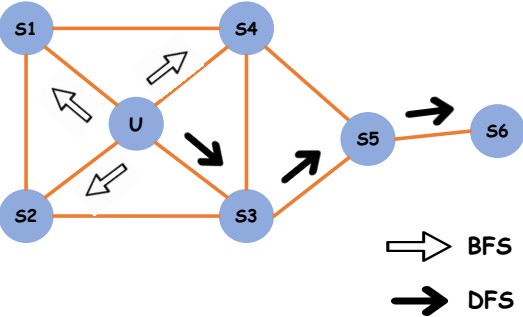

Figure 5: Demo of BFS and DFS dataset partition method.

## 6.2 The disscusion about the embedding replacement

As shown in Figure Figure 6, the model with expended vocabulary treats the entire protein ID as a whole when processing protein IDs. In contrast, the tokenizer of the model with original vocabulary will split the protein IDs. The split protein IDs will affect the model's performance in subsequent PPI tasks.

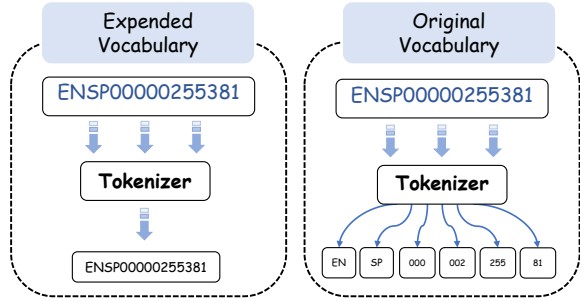

Figure 6: Expended vocabulary vs original vocabulary

## 6.3 The Datasets

### 6.3.1 Human dataset

The Human dataset contains 4577 unique proteins and 75875 interactions between proteins. The distribution of the Human dataset is shown on Table 4.

| Type | Count | Percentage |
|---|---|---|
| Binding | 17977 | 23.70% |
| Activation | 15470 | 20.41% |
| Catalysis | 11115 | 14.65% |
| Inhibition | 10611 | 13.99% |
| Expression | 9052 | 11.93% |
| Post-translational | 6255 | 8.25% |
| Modification and reaction | 5354 | 7.07% |
| Total Count | 75875 | |

Table 4: Distribution of Human dataset.

### 6.3.2 SHS27K, SHS148K and STRING

The STRING dataset is a large collection that contains 4,775,154 protein-protein interaction (PPI) records relevant to human biology, covering 15,335 distinct proteins and 572,568 unique interaction events. Two subsets of the STRING database are SHS27k and SHS148k. These subsets are curated by applying specific filters, such as selecting only proteins that are more than 50 amino acids long and exhibit less than 40% sequence similarity to each other, to ensure diversity and relevance. The SHS27k subset is smaller, with 16,912 PPI entries involving 1,690 proteins and a total of 63,408 interactions. The SHS148k subset is more extensive, containing 99,782 PPI entries, 5,189 proteins, and a high interaction count of 369,041. The distribution of the datasets is shown in Table 5

## 6.4 Implementation Details and Hyperparameters

ProLLM is trained on A40-48G. During training, the number of training epochs is 10, the learning rate is 3e-4, the per-device train batch size is 2, the per-device evaluation batch size is 2, the warmup steps are 400, and the weight decay is 0.01.

## 6.5 Detail in Embedding Replacement

To enhance the understanding of protein sequences, we adopt a method that integrates protein sequence vectorization with vocabulary expansion. First, we query the corresponding protein sequence $S_{P_{id}}$ based on the protein's unique identifier $P_{id}$ using the Ensemble

| Type | SHS27K | | SHS148K | | STRING | |
|------|--------|------------|--------|------------|--------|------------|
| | Count | Percentage | Count | Percentage | Count | Percentage |
| Reaction | 18,162 | 28.65% | 102,964 | 27.91% | 1,669,750 | 34.98% |
| Activation | 7,400 | 11.67% | 42,516 | 11.52% | 232,240 | 4.86% |
| Catalysis | 11,796 | 18.60% | 67,168 | 18.20% | 998,266 | 20.91% |
| Binding | 16,056 | 25.33% | 93,632 | 25.37% | 1,610,314 | 33.73% |
| Ptmod | 2,872 | 4.53% | 20,153 | 5.46% | 88,424 | 1.85% |
| Inhibition | 5,550 | 8.75% | 34,712 | 9.41% | 147,676 | 3.09% |
| Expression | 1,572 | 2.48% | 7,896 | 2.14% | 28,484 | 0.60% |

Table 5: Distribution of SHS27K, SHS148K, STRING dataset.

**Input template:**

```
This is the protein rational: The relationship between {{ENSP00000019103}}
and {{ENSP00000275216}} is catalysis, the relationship between
{{ENSP00000275216}} and {{ENSP00000267396}} is activation, the relationship
between {{ENSP00000267396}} and {{ENSP00000318944}} is binding, the relationship
between {{ENSP00000318944}} and {{ENSP00000275216}} is reaction. Let's think
step by step, what is relationship between {{ENSP00000019103}} and
{{ENSP00000275216}}?
```

**Target template:**

```
The relationship is activation.
```

Figure 7: One example of ProCoT prompt.

BioMart tool. Subsequently, the retrieved protein sequence $S_{P_{id}}$ is fed into the ProtTrans model, which outputs a $1 \times 1024$-dimensional vector $V_p$ encapsulating key information of the sequence. This vector is then used as the embedding vector for the new vocabulary item $P_{id}$ added to the Tokenizer's vocabulary. Through this approach, whenever the model encounters the identifier $P_{id}$, it utilizes the embedding vector $V_p$ generated by ProtTrans for processing, enabling the model to gain a deeper understanding of protein sequences.

## 6.6 Prompt in ProLLM

The prompt in ProLLM has two type: ProCot prompt and Instruction finetuning prompt. Figure 7 is an example prompt of ProCot. Figure 8 is the prompt of instruction fine-tuning.

**Raw data:**

**Instruction *I*:** Analyze this protein sequence and, based on conserved
domains or motifs, deduce its possible cellular function(s).
**Sequence *S*:** MRLRKKWWARPEMEASPLCIV…
**Metadata *M*:** tRNA (guanine-N7-)-methyltransferase activity…
**Output *O*:** Upon evaluating the structure of the protein with sequence, it
can be predicted that its biological function is primarily associated with
tRNA (guanine-N7-)-methyltransferase activity.

**Input template:**

Follow the instruction *I* to answer the question. The protein sequence is *S*.
*M* is the information about subcellular localization, primary function, and
biological process.

**Target template:**

The answer is : *O*

Figure 8: Example of instruction fine-tuning prompt.

