# OpenReview forum: "ProLLM: Protein Chain-of-Thoughts Enhanced LLM for Protein-Protein Interaction Prediction"
_colmweb.org/COLM/2024/Conference — COLM_

### Official Review · Reviewer_YsHM · 2024-04-21

**Rating:** 5
**Confidence:** 4
**Ethics Flag:** 1

**Summary:**

The paper introduces ProLLM, a new framework employing a LLM specifically tailored for predicting Protein-Protein Interactions (PPIs), crucial in understanding biological functions and diseases. It leverages a unique approach called Protein Chain-of-Thought (ProCoT), which simulates signaling pathways in natural language prompts to predict interactions, focusing on both physical and non-physical connections between proteins. ProLLM outperforms traditional graph-based methods and existing LLM approaches by incorporating protein-specific embeddings and instruction fine-tuning.

**Questions To Authors:**

See above

**Reasons To Accept:**

1. The paper introduces an innovative approach to solving PPI task, which improves prediction accuracy and meanwhile enhances the interpretability.

2. The authors provide a comprehensive survey in the Related Work section, detailing previous studies and advancements.

3. The overall presentation of the paper is of high quality. The text is well-crafted and fluid. The figures are clear and effective.

**Reasons To Reject:**

1. My primary concern lies with the practice in this paper behind flattening highly structured protein data into sequential language. The authors mention that "we directly integrate the protein vectors generated from ProtTrans into our model to replace the original word embedding in the protein name’s place", which describes their method of encoding protein data for the LLM. However, the paper lacks essential details and descriptions of how this protein encoder aligns with the backbone LLM. The specifics of how alignment training is conducted are also not clear.


2. The experimental analysis appears to be the weakest aspect of the paper. The analysis conducted is neither thorough nor solid. The authors do not delve deeply into exploring how their proposed ProCoT actually functions, nor do they examine scenarios where it might fail. There is a need for well-designed experiments to validate the mechanisms described in section 3.1.3, which would significantly strengthen the conclusions and its robustness.

---

> ### Author Rebuttal · Authors · 2024-05-30
>
> **Question 1: My primary concern lies with the practice in this paper behind flattening highly structured protein data into sequential language. And the paper needs descriptions of how this embedding of ProTrans align with the natural language embedding of Flan-T5-large.**
>
> **Answer:**
>
> Firstly, besides the sequential language we used for PPI prediction, the 'ProTrans' [1] embedding we incorporate includes protein structural information.
>
> Since ProTrans is trained on protein sequences based on T5, its embeddings share similar embedding spaces with Flan-T5-large due to their common architecture and training paradigms. Therefore, ProTrans naturally aligns well with our ProLLM, which is based on the Flan-T5 model. Additionally, during the fine-tuning process of ProLLM, the embeddings of ProTrans will be updated and aligned with the embeddings of natural language to predict protein interactions. Table 2 in section 4.3 demonstrates that ProLLM based on Flan-t5-large has better alignment performance for ProTrans compared to other backbones.
>
> We will re-emphasize the above descriptions and the results in Table 2 in the manuscript so as to clarify how the protein encoder integrates with the backbone LLM during fine-tuning and alignment.
>
> [1] Elnaggar. ProtTrans: Toward Understanding the Language of Life Through Self-Supervised Learning. IEEE TPAMI. (2021).
>
> ---
>
> **Question 2: The authors could further explore how the proposed ProCoT functions and examine potential failure scenarios. Implementing well-designed experiments to validate the mechanisms described in section 3.1.3 would significantly enhance the paper's conclusions and robustness.**
>
> As discussed in section 3.1.3, ProCoT functions by representing signaling pathways as chain-of-thought reasoning rules. This is because in biological systems, signals between proteins are transmitted through signaling pathways, facilitating protein-protein interactions. ProCoT can embed these signaling pathways into a chain input for LLM, enabling it to repeatedly utilize these pathways to predict PPIs. We further conducted experiments to validate this. In Table 3 of Section 4.4, we disrupt the sequence of protein signals in ProCoT, which causes a significant drop in prediction accuracy. This is consistent with the working mechanism of ProCoT. Thank you for your feedback. We will re-emphasize the results of Section 4.4 Table 3 in Section 3.1.3 so as to allow readers to gain a deeper understanding of ProCoT.

---

> > ### Author Response · Authors · 2024-06-06
> > **A Friendly Reminder**
> >
> > Dear Reviewer YsHM,
> >
> > Thanks again for your valuable comments and precious time. As the author-reviewer discussion period draws to a close, we genuinely hope you could have a look at the new results and clarifications and kindly let us know if they have addressed your concerns. We would appreciate the opportunity to engage further if needed.
> >
> > Best,
> >
> > Authors of Paper 51

---

### Official Review · Reviewer_bZnW · 2024-05-08

**Rating:** 6
**Confidence:** 4
**Ethics Flag:** 1

**Summary:**

- The paper presents a sophisticated approach that leverages large language models (LLMs) to predict protein-protein interactions (PPIs) by representing the complex signaling pathways in a natural language format. The use of the Protein Chain of Thought (ProCoT) to simulate signaling pathways as a reasoning chain is a unique method that effectively blends computational biology with advanced NLP techniques.
- The paper is well-structured, methodologically sound, and systematically presented. The inclusion of detailed figures, such as the ProLLM framework and examples of ProCoT, enhances the clarity and understandability of the proposed method.
- The concept of using LLMs, specifically the adapted Flan-T5 model infused with protein-specific embeddings, for modeling PPIs through natural language is novel and innovative. The introduction of ProCoT as a means to contextualize protein interactions within signaling pathways demonstrates original thinking.
- The significance of this work is high, as it proposes a new avenue for PPI prediction which is crucial for understanding biological functions and the mechanism of diseases. The results show a significant improvement over existing methods, which could influence future research in computational biology and drug development.

**Questions To Authors:**

- Can you elaborate on how ProLLM handles the variability and potential noise in real-world biological datasets?
- How does ProLLM ensure that the model does not overfit to the training data, given the complexity of the model and the specificity of the datasets used?
- Given the innovative use of language models in predicting PPIs, what are the limitations of this approach in terms of computational resources and scalability?

**Reasons To Accept:**

- Innovative Conceptual Framework: The paper introduces an innovative method of simulating protein signaling pathways using natural language processing models. By transforming protein interactions into a chain of reasoning with the ProCoT format, the paper leverages the power of LLMs, such as Flan-T5, in a novel biological context. This approach is pioneering as it merges two distinct fields: computational biology and language modeling. This methodology could lead to more intuitive and explainable models for protein interaction, which are crucial for understanding complex biological processes and drug discovery.
- Impressive Performance: The model demonstrates superior performance metrics, particularly in terms of prediction accuracy and generalizability across multiple datasets, as indicated by the micro-F1 scores provided. The performance improvements are robust, with ProLLM outperforming existing graph-based and LLM-based methods significantly. The high performance indicates that ProLLM can be a more effective tool in the predictive analysis of PPIs, which could transform how researchers approach the prediction of biological interactions.
- Potential for Broad Impact: The model's ability to accurately predict interactions can directly contribute to better understanding of diseases and the development of therapeutic strategies, especially in complex diseases like cancer where protein interactions play a crucial role. Beyond immediate research applications, the methodology could be adapted for other types of molecular or genetic interactions, broadening its impact.
- Well-Documented and Reproducible Research: The authors provide access to source code and detailed documentation, which facilitates replication and further investigation by other researchers. This openness not only enhances the scientific integrity of the research but also encourages further innovations and improvements by the broader scientific community.

**Reasons To Reject:**

- Scope of Evaluation and Applicability: While the paper presents strong performance on benchmark datasets, these datasets often have well-defined and controlled conditions that may not adequately represent the messy, heterogeneous data encountered in real-world biological research. This raises concerns about the model's robustness and effectiveness outside of academic test conditions. The focus on specific datasets might have tailored the model excessively to those conditions. There is insufficient evidence on how well the model adapts to datasets with different characteristics or to PPI predictions in less-studied organisms, which limits the understanding of its broader applicability.
- Practical Implications for Broader Adoption: The computational demands for training and running such large-scale models, especially when fine-tuning with extensive protein datasets, might be prohibitive for many research institutions, particularly those with limited IT infrastructure. The dynamic nature of both biological research (with continually updating data and new protein interactions being discovered) and machine learning models (requiring regular updates to maintain performance) poses significant challenges in maintaining the relevance and accuracy of the model over time.
- Potential Overfitting and Model Bias: The use of a highly complex model architecture, such as a large language model tailored with specific protein data, increases the risk of overfitting. This concern is exacerbated if the model has been overly optimized for high performance on the benchmark datasets without sufficient regularization or cross-validation strategies. The adaptation of language models to biological data involves assumptions and simplifications that may introduce biases, particularly in how protein interactions are represented and predicted. These biases could affect the predictive accuracy when the model is applied to new or different types of protein data, leading to erroneous conclusions or missed interactions.

---

> ### Author Rebuttal · Authors · 2024-05-30
>
> **Question 1: Can you elaborate on how ProLLM handles the variability and potential noise in real-world biological datasets?**
>
> **Answer:**
>
> ProLLM predicts protein-protein interactions (PPI) through protein signaling pathways, which are commonly present in all proteins and are mostly decided by the fundamental structure of the protein. As shown in Table 1, ProLLM maintains high performance even when encountering proteins not present in the training set. This indicates that our model has learned the patterns of signaling pathways between proteins, demonstrating its wide applicability.
>
> ---
>
> **Question 2: How does ProLLM ensure that the model does not overfit to the training data, given the complexity of the model and the specificity of the datasets used?**
>
> **Answer:**
>
> ProLLM learns the signaling pathways between proteins rather than specific patterns in the training data, which helps our ProLLM avoid overfitting. Table 1 demonstrates that our model has learned the signaling pathway patterns because it performs efficiently on unseen datasets, further evidencing that our model does not suffer from overfitting. Additionally, we used a validation set to monitor the training process of the model. The average loss on the validation sets for the Human, SHS27K, SHS148K, and STRING datasets was 0.0026, 0.0024, 0.0023, and 0.0021, respectively. This indicates that the model does not exhibit signs of overfitting.
>
>
>
> |         | SHS27K       | SHS148K      | STRING       |
> |---------|--------------|--------------|--------------|
> | micro-F1 | 78.54±4.21   | 81.27±2.04   | 82.59±1.95   |
>
> Table 1: The Performance of ProLLM in different unseen datasets; the ProLLM is only trained on Human proteins.
>
>
>
>
> ---
>
> **Question 3: Given the innovative use of language models in predicting PPIs, what are the limitations of this approach in terms of computational resources and scalability?**
>
> **Answer:**
>
> Our inference and training can be accomplished on standard A40 GPUs, indicating that our model does not require high-end hardware. We will include these details about computational resources in the manuscript to help readers better understand our method's computational resource requirements and scalability.

---

> > ### Author Response · Authors · 2024-06-06
> > **A Friendly Reminder**
> >
> > Dear Reviewer bZnW,
> >
> > Thanks again for your valuable comments and precious time. As the author-reviewer discussion period draws to a close, we genuinely hope you could have a look at the new results and clarifications and kindly let us know if they have addressed your concerns. We would appreciate the opportunity to engage further if needed.
> >
> > Best,
> >
> > Authors of Paper 51

---

### Official Review · Reviewer_F5aN · 2024-05-13

**Rating:** 8
**Confidence:** 4
**Ethics Flag:** 1

**Summary:**

This manuscript introduces an approach for predicting protein-protein interactions, especially those due to indirect interactions, such as those found in a pathway. The approach uses an LLM that has been upgraded with protein vectors, fine-tuned for protein tasks, and - most importantly - set up using chain-of-thought to reason through the process step by step.

**Questions To Authors:**

Given the strong performance, the ProTrans embeddings clearly work well with the other embeddings used. However different training processes typically do not produce embeddings that align in this way. How was this accomplished?

**Reasons To Accept:**

Protein-protein interaction prediction is a critical task for understanding biological systems, with applications across all of biology and medicine. This manuscript is well-written, with a discussion of related work that appears to be comprehensive, and with a very comprehensive evaluation and results that clearly support the methods proposed. Moreover, as stated by the manuscript, this may be the first work to explore protein-protein interaction as a natural language inference problem.

**Reasons To Reject:**

The only concerns I have are quite minor:

1. It would be helpful to explain a bit better why indirect protein-protein interactions are so important and difficult.

2. The manuscript refers to the approach as converting the PPI prediction into a natural language processing problem. While quite possibly true, natural language processing is a very large field. I believe that the task here would be more specifically termed natural language inference or, possibly, natural language understanding.

3. Section 3.1.2 references the principles of "co-expression" and "co-localization" but these are not defined.

And a couple of edits:
1. I think "learn the law of signal pathways" (page 2) would be more accurate as "learn the mechanism of signal pathways."

2. In section 4.2 "Procot" should probably by "ProCoT"

---

> ### Author Rebuttal · Authors · 2024-05-30
>
> **Q1: Why are indirect protein-protein interactions crucial and challenging to understand?**
>
> In living organisms, signals are transmitted through pathways where upstream proteins send biological signals to downstream proteins via intermediary signal molecule proteins [1]. These indirect protein-protein interactions are crucial for essential life processes, making their understanding vital.
>
> Signal pathways involve multiple intermediary proteins, resulting in complexity and heterogeneity. To address this, we propose leveraging large language models' reasoning capabilities, specifically the Chain of Thought method [2], to solve signal pathway problems with ProCoT.
>
> [1] Anti-cancer effects of Polyphyllin I: An update in 5 years. *Chemico-Biological Interactions*.
>
> [2] Chain-of-thought prompting elicits reasoning in large language models. *NeurIPS*.
>
> **Q2+Q4: Re-wording some technical terms**
>
> We will describe our approach as natural language inference, which is more specific. Additionally, we will correct 'learn the law of signal pathways' to 'learn the mechanism of signal pathways.'
>
> **Q3: Section 3.1.2 references the principles of "co-expression" and "co-localization" but these are not defined**
>
> Co-expression means that proteins expressed together are likely to interact; co-localization means that proteins in the same subcellular area have a higher chance of interaction [3]. We will add the descriptions to the paper.
>
> [3] Deciphering cell-cell interactions and communication from gene expression. *Nat Rev Genet*.
>
> **Q5: How does ProTran align well with ProLLM despite different training processes?**
>
> It is noteworthy that the ProTrans embeddings and other embeddings used in our project align strongly. ProTrans, trained using the T5 model, naturally aligns with our ProLLM which is also based on T5. Both models share similar embedding spaces due to their common architecture and training paradigms. Additionally, fine-tuning ProLLM ensures that ProTrans embeddings and natural language tokens align with each other.
>
> If we replace our backbone with a different model, such as LLaMA, we must fine-tune the ProTrans embeddings or use alternative embeddings compatible with the new model's embedding space. As shown in Section 4.3 Table 2, where we experiment with different backbones (Flan-T5 and LLaMA), ProLLM based on Flan-T5-large performs better due to the shared embedding space with ProTrans. Thanks for the suggestion and we will re-emphasize this in the paper.

---

> > ### Comment · Reviewer_F5aN · 2024-05-31
> >
> > Thank you for the response. These all make sense and I assume there will be (very minor) updates to the manuscript. My (positive) recommendation remains unchanged.

---

> > > ### Author Response · Authors · 2024-06-01
> > >
> > > Thank you very much for your support and feedback. We will add the new clarification in the revised version.

---

### Official Review · Reviewer_rKT9 · 2024-05-23

**Rating:** 7
**Confidence:** 3
**Ethics Flag:** 1

**Summary:**

The authors introduce a novel framework, ProLLM that uses LLMs to investigate PPIs. For this, they devise ProCoT that transforms the gene signaling pathways and protein functions into NL prompts. They discuss how to train and fine tune ProCoT. Experiments and ablation studies are provided that demonstrate the robustness of ProLLM.

The paper is well written, and previous literature and gaps in the field are thoroughly discussed.

**Questions To Authors:**

* Related to Definition 2 and thereafter: How can prompts handle the diverse and various nature of feedback/feedforward loops amongst the genes/proteins? For example, many of these connections are triggered by environmental cues. While it is impossible to know all/what the environmental contexts are, how are the authors making sure that, at least, the canonical pathway directions are captured sans context?
* Section 3.1.2. Expand DFS in the main text (and then refer to it in the appendix)

**Reasons To Accept:**

The proposed method seems to be effective; the work shows the importance of using an LLM for PPI for the first time.

**Reasons To Reject:**

None

---

> ### Author Rebuttal · Authors · 2024-05-30
>
> **Question 1: How do prompts ensure canonical pathway directions among proteins are captured despite diverse feedback/feedforward loops and unknown environmental contexts?**
>
> **Answer:**
>
> Protein-protein interactions are primarily determined by the intrinsic properties of the proteins themselves, such as protein structure, and are largely independent of external environmental factors [1]. The records on protein-protein interaction data included in our databases are supported by wet experimental data and are highly credible. This ensures the high quality of our training data. It is important to note that the primary factors influencing protein interactions include the proteins' expression level and subcellular localization [3]. For instance, if two proteins are expressed concurrently in the same cell and are proximal within the cellular space, they are more likely to interact. Conversely, distant proteins are less likely to interact [4]. This phenomenon allows our ProCoT to accurately reflect classic biological pathways well recognized within the scientific community and have significant implications in disease treatment and drug development [5].
>
> [1] Pawson, T. Protein modules and signaling networks. *Nature* 1995.
>
> [2] Szklarczyk D. The STRING database in 2023: protein-protein association networks and functional enrichment analyses for any sequenced genome of interest. *Nucleic Acids Res.* 2023
>
> [3] Villanueva, E., Smith, T., Pizzinga, M. et al. System-wide analysis of RNA and protein subcellular localization dynamics. *Nat Methods* 2024.
>
> [4] Bickmore WA. Addressing protein localization within the nucleus. *EMBO J.* 2002.
>
> [5] Pham M. Discovery of disease- and drug-specific pathways through community structures of a literature network. *Bioinformatics* 2020.
>
>
> **Question 2: Section 3.1.2. Expand DFS in the main text (and then refer to it in the appendix)**
>
> **Answer:**
>
> Thank you for your suggestion. We will expand the explanation of depth first search (DFS) in Section 3.1.2 by including a brief description, and then refer readers to Section 6.1 in the Appendix for more detailed information. This revision will be incorporated in the updated version of the paper.

---

### Decision · Program_Chairs · 2024-07-10

**Decision:**

Accept

**Comment:**

Submission 51, titled "ProLLM: Leveraging Large Language Models for Predicting Protein-Protein Interactions," introduces an innovative framework that utilizes large language models (LLMs) to investigate protein-protein interactions (PPIs). The authors present ProCoT, a novel method that transforms gene signaling pathways and protein functions into natural language prompts, allowing for fine-tuning and training to predict PPIs. The submission has been reviewed by four reviewers, all of whom recognize the novelty and potential impact of the proposed method.

Based on the reviewers' feedback and the authors' thorough rebuttals, Submission 51 presents a highly innovative approach with the potential to significantly advance the field of PPI prediction. While there are some concerns regarding the generalizability and detailed experimental validation, the overall contribution, novelty, and potential impact of the work are substantial. The authors have shown a commitment to addressing the concerns raised, which further strengthens the case for acceptance. The authors should ensure that the final manuscript includes the clarifications and additional experimental results promised in their rebuttals, particularly addressing the alignment of embeddings and the robustness of the model on diverse datasets.